# Mutual Influence of Psoriasis and Sport

**DOI:** 10.3390/medicina57020161

**Published:** 2021-02-10

**Authors:** Paolo Custurone, Laura Macca, Lucrezia Bertino, Debora Di Mauro, Fabio Trimarchi, Mario Vaccaro, Francesco Borgia

**Affiliations:** 1Section of Dermatology, Department of Clinical and Experimental Medicine, University of Messina, 98125 Messina, Italy; paolo.custurone@gmail.com (P.C.); lauramacca7@gmail.com (L.M.); bertino.lucrezia@gmail.com (L.B.); mario.vaccaro@unime.it (M.V.); 2Department of Biomedical, Dental Sciences and Morphofunctional Imaging, AOU “G. Martino”, 98125 Messina, Italy; debora.dimauro@unime.it (D.D.M.); fabio.trimarchi@unime.it (F.T.)

**Keywords:** psoriasis, sport, physical activity, exercise, metabolic risk, diet, psoriatic arthritis

## Abstract

The link between psoriasis and sport is a controversial issue. The topic has been poorly investigated, and nowadays there are many unsolved questions, dealing with the role of psoriasis in influencing the sporting habits of patients and, vice versa, the impact of sport activity on course, severity and extent of the disease, with particular regard to the indirect benefits on cardiovascular risk and metabolic syndrome. Moreover, the role of physical activity on patients’ quality of life and the potential limitations on physical activity due to joint involvement have not been well elucidated until now. In this narrative review we will try to provide answers to these queries.

## 1. Introduction

Psoriasis (Pso) is a chronic inflammatory immune-mediated skin disease with an estimated worldwide prevalence of approximately 2–3% [1]. For a long time considered to be an exclusively cutaneous disorder, Pso is now recognized as a systemic inflammatory disorder sharing pathogenic pathways with many other chronic and progressive health diseases, including psoriatic arthritis, metabolic syndrome (MetS), cardio-vascular disorders, inflammatory bowel disease, uveitis, obstructive sleep apnea, non-alcoholic fatty liver disease (NAFLD), psychiatric conditions, hypoacusia [2,3,4]. In particular, Pso patients are more frequently overweight or obese than the general population, and the severity of Pso correlates to body mass index (BMI) [1]. The chronic course of the disease makes patients conscious that they will have to deal with this condition for the rest of their life, with a detrimental effect on every-day life. Skin involvement causes a significant physical and psychosocial burden. During life course, patients experience feelings of anger, frustration, helplessness, embarrassment, self-consciousness. The disease also dramatically impacts on sexual behavior and work productivity. Musculoskeletal involvement, presenting as peripheral arthritis, dactylitis, enthesitis and spondylitis, may further aggravate the prognosis of the patients, adding functional limitations to daily activities. The link between Pso and sport is a controversial issue. Physical activity has been associated to Pso in a negative way, even if recent studies have highlighted the potential beneficial effect on the natural course of the disease and, as consequence, on patients’ quality of life (QoL). The topic has been poorly investigated, and nowadays there are many unsolved questions, dealing with the role of the skin disease in influencing the sporting habits of patients and, vice versa, the impact of sport activity on course, severity and extent of the disease, with particular regard to the indirect benefits on cardiovascular risk and metabolic syndrome potentially derived from practicing sport in a Pso context. Moreover, the role of physical activity on patients’ QoL and the potential limitations on physical activity due to joint involvement have not been well elucidated until now. In this narrative review we will try to provide answers to these queries.

## 2. Materials and Methods

This research was conducted using the PubMed database (https://ncbi.nlm.nih.gov/PubMed accessed on 30 December 2020), using the keys related to psoriasis and sport. The search string used was “psoriasis”[All Fields] AND (“physical exercise”[All Fields] OR “physical activity”[All Fields] OR “sport”[All Fields].

Using the PRISMA guidelines for the drafting of the review, we read the abstract of each article whose title suggested the association between psoriasis and sport. The entire article was read only if the abstract indicated that the article potentially met our inclusion criteria: English language, research papers, and studies on human population only. Finally, the references of the articles selected were examined in order to identify further studies that could be included in the review, based on the same criteria. Papers identified from the title, abstract, or full text as irrelevant to the topic in question, as well as those potentially relevant but performed on animals or cells, were excluded.

Given the heterogeneity of the studies examined and the small number of results, we chose not to carry out a systematic review of the literature but a more discursive review.

## 3. Results

The initial PubMed search yielded 72 articles. Of these, 43 were not considered because the title and/or abstract suggested that they were not research papers, not written in English, not performed on human populations only, not relevant to the outcome of interest, and/or because the full text was not available.

The 29 studies selected for inclusion in the review were sorted in three subcategories, accompanied by a resume template each, namely The impact of Sports on Psoriasis (Table 1), The Relevance of Sports to Comorbidities of Psoriasis (Table 2) and The Effect of Psoriasis on the Sport Activities (Table 3). Author(s) and year of publication, characteristics of the population studied (adults, children or both), number of patients and controls, any scores or factors evaluated because of their possible link with the object of the review, and the main outcome(s) of each study were included in the tables.

### 3.1. The Impact of Sports on Psoriasis

Sporting activity has several effects on the Pso patients: it can ameliorate the lesions, leave them as they are or worsen them. The following papers deal about weight loss and physical activity and their role in influencing the severity of psoriasis.

Schwarz et al. conducted a study, called ERAPSO, on a population of 9940 german patients affected by both Pso and overweight. Most of them had attempted an approach to weight loss through diet and sports. About 20% were exercising during the study in order to lose weight. Patients with a BMI greater than 25 kg/m^2^ reported greater discomfort while practicing sports than those with a lower BMI. This percentage grew in patients with moderate to severe forms of psoriasis, especially if they practiced physical activity without constancy. Most participants (90.2%) with BMI ≥ 25 kg/m^2^ perceived themselves as overweight. Skin involvement was considered a limit to perform sport, leading to complete abandonment of any physical activities, emphasizing the cosmetic aspect of the disease. This in spite of the positive influence of sport on Pso severity. Greater benefits were reported from activities such as cycling and walking compared to endurance sports, and even more so if compared to home chores. Furthermore, it was noted that patients with moderate or severe psoriasis were more sensitive to positive or negative feedback from the doctor about food and sports habits. Among the conclusions of the study, it is suggested that the optimal pathology control could facilitate weight loss through physical activity [5].

In another study, Frankel et al. investigated motor activity habits in US nurses diagnosed with psoriasis through questionnaires specifically administered to evaluate the correlation between vigorous exercise and risk of developing Pso. They found an inverse relationship between the two aspects. Although not all grades of physical activity (calculated with Metabolic Equivalent Task (MET)/hour per week) have shown statistically significant results, the benefits resulting from vigorous exercise seem to suggest a threshold effect. Furthermore, the variable “amount of exercise” appeared independently related to the risk of developing psoriasis with respect to the BMI level of the patients. The “vigorous” activity was set as 105′/week of running or 180′/week of swimming/tennis, while mere walking did not produce positive or negative implications for the disease activity. Only certain aerobic activities, such as running, have proven useful in preventing psoriasis. It may be due to the fact that other sports activities with similar energy expense showed a discontinuous level of intensity. Moreover, it was very rare to observe patients engaged in a single high intensity activity for prolonged periods of time rather than various activities with variable intensity and shorter periods of time. Two main hypotheses have been proposed to explain such results: modification of the cytokine profile from a pro-inflammatory to a less pro-inflammatory subset (IL-6, TNF-alpha, leptin and C-reactive protein) and reduction of psycho-emotional stress which, through a modulation of the activity of T lymphocytes, would regulate the disease’s activity. A possible limitation to the study could be represented by the confounding element of the beneficial action of the sun carried out by ultraviolet (UV) rays; however, patients who spent the same amount of energy walking outdoors compared to those who did vigorous activity outdoors reported significantly less marked results [6].

Do et al. researched the link between the extent of psoriatic lesions and the likelihood of participating in leisure-time moderate to vigorous physical activity (MVPA). The MET-minutes spent during MVPA among the patients was also evaluated. They found out that, even though the difference in the last 30 days of the time doing sports was not statistically significant between the two groups, the METs consumed per minute were 30% less in the psoriatic group. This finding was attributed, by the authors, to the fact that patients with more extensive skin lesions are less prone in engaging physical activity [7].

In the large study of Balato et al. the relationship between sport and Pso was investigated comparing 416 sportsmen, 489 controls and 400 psoriatic patients (sex- and age-matched). The authors noted that although there was a similar percentage of family history of psoriasis among sportsmen and controls, the prevalence of the disease was significantly lower in the sportsmen group. Only 1.7% of sportspeople had Pso compared to 5.4% of controls. The number of subjects who actively played sports was much less among the Pso patients than the controls, suggesting that Pso may be a limitation to physical activity. Among psoriatic patients, a significant number of them (95) practiced sports before the disease onset. The Authors suggest that physical activity would exert positive effects on Pso not only through a simple mechanism of weight loss but also at a molecular level, inducing epigenetic modifications (DNA methylation and histone acetylation) and decreasing pro-inflammatory cytokines (TNF-alpha, IL-6) and leptin. The main limitations of this study are that the administered questionnaires were doctor specific and not based on those normally used (Body Surface Area, BSA; Psoriasis Area Severity Index, PASI; Dermatology Life Quality Index, DLQI) and that, to better define the link between sport and Pso, a larger sample of the population would be requested [8].

Megna et al. compared second-league footballers and healthy controls with overlapping physical characteristics (age, gender and BMI) and similar family history for psoriasis by means of a questionnaire. It emerged that players affected by psoriasis did not perceive sport as a factor negatively influencing the natural history of their disease. A higher prevalence of psoriasis was found in controls who did not perform any physical activity with respect to those regularly involved in exercise programs, suggesting that physical activity may play a protective role against the development of cutaneous manifestations of the disease as well as its cardio-metabolic comorbidities [9].

A relevant question about the possible side effects of sport in case of psoriatic disease was posed by Brophy et al. In a case report, they describe the case of a 39-year-old athlete who, presenting with recurrent monolateral knee effusions, received a late diagnosis of psoriatic arthritis since his injuries were initially diagnosed as due to trauma while playing; in addition, the patient had a positive family history of seronegative and non-seronegative spondyloarthritis. Following a suitable treatment, the patient not just recovered the complete function of his knee, but also noticed improvements in other joints which previously had not prevented him from attending competitive sports. As the Authors say, a patient suffering from oligoarticular forms of psoriatic arthritis not only would benefit from an early and aggressive treatment, but also should be aware of an intrinsic structural weakness of his own cartilages, and therefore limit the use of the joints in sports like the contact ones. We see, therefore, how sport presents itself both as a potential target for the treatment of psoriatic disease and, on the other hand, as a source of possible injury [10].

A similar event was reported by Corn et al., concerning a patient known for a history of psoriasis who had seen the appearance of characteristic lesions on the soles of the feet following sports, seeing how the Koebner phenomenon could, therefore, be a spy of a pathology otherwise with minimal clinical manifestations [11].

### 3.2. The Relevance of Sports to Comorbidities of Psoriasis

Psoriasis can have several comorbidities linked to it, mostly dismetabolic ones caused by lack of exercise. One of the main means to evaluate a patient’s fitness is through examination of the cardiorespiratory fitness (CRF) The link between CRF and the severity of psoriasis was highlighted by Wilson et al. In this study, important results and questions emerge regarding the role of physiological processes that occur during exercise (sweating and heat dissipation) in relation to psoriatic plaques. Fitness measurement was performed by running tests on the treadmill. The results demonstrated a significant decrease in cardiorespiratory fitness in psoriatic patients compared to healthy controls, suggesting other mechanisms besides the lack of constant training. According to the Authors the disease activity, measured through the PASI index, can also be related to heart rate, demonstrating a correlation between psoriasis and cardio-metabolic risk. It should be noted that cardiorespiratory fitness was not investigated in this study through maximal exercises that could better identify differences between the groups. Moreover, the cohort of partecipants is too small to clearly demonstrate the association between disease severity and CRF lowering. However, this study offers interesting research ideas regarding factors potentially related to psoriatic disease and their role in decreasing CRF in such patients [12].

In another study, Wilson proposed a specific questionnaire (global physical activity questionnaire, GPAQ) to know what kind of activities the psoriatic patient engage in order to lose weight. The study population included 9174 participants, of which 232 had psoriasis. The results showed that only 48% of the Pso patients tried to lose weight through physical activity, compared to the 62.4% of healthy subjects. Moreover, just 16.1% of Pso patients performed vigorous physical activity, compared to 28% of unaffected participants. It is the author’s opinion that patients affected by both psoriasis and obesity might practice sport less than patients affected by just one of these conditions. A better dialog between physician and patients could help them to find new ways of doing sports in order to lose weight [13].

On the same note, Naldi et al. conducted a study aimed at finding a correlation between psoriasis and its improvement through diet and physical excercise. In this trial, 303 overweight-obese patients were enrolled, and it was proposed to them either a 15 min meeting in which they were instructed what kind of behaviours they should have adopted in order to lose weight or a full dietary/physical programme, managed by a physician. We believe that this study, even though claims overall significant results, presents several limitations: the patients treated with systemic drug were all put together without any further selection criterion (e.g., dividing them by biologics, cyclosporine, acitretine etc.); a closer supervision in the interventional arm could have produced intrinsically better results; this study was conducted in a short period of time, without a proper follow-up to assess the maintenance of this loss of weight through time. Overall, this study, once again, tells us how much it is important for a complete medical team to follow through the years psoriatic patients in order to achieve complete control over this disease [14].

The studies cited so far are mainly based on the relationship between sport and psoriasis; it must be said, however, that even simple weight loss could be a sufficient factor to determine a clinical improvement. In a large study, Kim et al. showed that the metabolic syndrome features and Pso lesions are directly related [15]. Moreover, Gyldenløve et al. lead an interesting study about insulin resistance and risk of psoriasis. The Authors conclusion, even if based on a small sample, was that psoriasis may represent a prediabetic condition as a result of a common genetic background or a common pathogenetic pattern of pro-inflammatory cytokines [16].

Jensen et al. have conducted a study on long-term efficacy of weight loss in Pso patients. Even though in this case the element “sport activity” has not been taken into consideration for weight loss purposes, it is undeniable that slimming is a complex process able to exert, if supported by regular sporting activity, improvement of PASI and DLQI [17]. As reviewed by other authors, a weight loss of 20 kg in Pso patients can lead to a significant improvement of DLQI score and weight management. If the endpoint of 20 kg is reached, the weight loss can be maintained through time more easily. It is the author’s opinion that, given the significant improvement of the skin condition and the cardiovascular risk associated with excessive weight, further studies are necessary to validate the diet formula as an adjuvant treatment for psoriasis [18].

Out of all the possible alimentary regimens, the mediterrean diet seems to be the one less linked to severe forms of psoriasis, thus suggesting its protective role in the progression of this disease [19]. The poor control over their diet and sporting activity can determine long-term higher intake of antipsoriatic drugs and possibly more side effects linked to the dismetabolic status [20].

Finally, another study by Ferguson et al. suggests a role of central adiposity in the development of psoriasis, psoriatic arthritis and rheumatoid arthritis: all of these conditions were significantly associated with overweight and lack of physical activity, even after correction of the BMI, thus suggesting that the weight could be somehow linked to the development of several autoimmune diseases [21].

### 3.3. The Effect of Psoriasis on the Sport Activities

It is well-known that psoriasis has a strong impact on patients’ life. It influences working habits, poses a significant financial burden, but most of all, dramatically impairs their quality of life and psychological status. The visibility of the disease is a deterrent for the practice of some sports, especially for those requiring the exposure of large parts of the body. This has been well documented in the study of Ramsay et al., in which 72% of patients declared to consciously avoid swimming, a percentage that went down to a still consistent 46% even in the case of mild psoriasis. Similarly, 64% of patients avoided bath and showers in public. Pso also influenced other aspects linked to sport activity like the choice of color and types of clothing (in particular summer clothes). Over half of the patients reported the sensation that their medical condition could represent a social stigma, as they are perceived to be contagious, causing social phobia and refraining from activities outside the house. Furthermore, asking the patients what the worst factors of the disease were, they replied for about 80% that one’s appearance, linked to the disease and the itch, represented a major limit to carry out some types of sport. This observation could suggest that, if these patients have had pleasure in practicing contact sports, psoriasis would prove to be an insurmountable psychological obstacle regardless of the severity of the disease [22].

On a group of patients (223) ranged 18 to 83 years of age, Nyunt et al. used the classic DLQI to assess the impact on the quality of life by psoriasis. Factors significantly associated with scarce quality of life included sports, PASI, marital status, level of work or study, nail dystrophy, visible affected areas, itching, sleep quality, stress and infections. The Authors suggest that Pso would invalidate sports activity for psychological and social reasons (shame of one’s own body) or for the exposure to repeated physical trauma as important stimulus for the appearance of Koebner phenomenon. The main limits of this study are the scarce variety of cases, since it was conducted in a tertiary medical center of and the pool of individuals was composed of patients with severe psoriasis. Furthermore, the calculation of BSA and PASI of skin lesions was carried out without taking precise measurements but “at a glance” [23].

About the relevance of the research items of the DLQI questionnaire, Rencz et al. observed that some of these, such as sport (28.4%) and others (sexual difficulty and study/work), are considered as irrelevant, especially by a portion of the population made up of elderly patients/female patients/scarcely educated patients. The Authors suggest reconsidering the period of time for evaluating the questionnaire in order to highlight aspects previously considered unnecessary by patients. The conclusion of the study is controversial since, if on one hand increasing the observational period (one week currently) would increase the relevance of the item, this would require at the same time a greater memory effort. The Authors conclude that, however, the DLQI is a valid and necessary tool for defining the quality of life of Pso patients, in relation to the increase in average age [24].

Barbieri et al. investigated the impact of socioeconomic factors on “not relevant” responses (NRRs) on the DLQI. They found that the items regarding sport, sexual difficulties and work/study had the greatest frequency of NRRs, especially in unemployed people or on the basis of marital status. The Authors concluded that differences of socioeconomic status may cause underestimation of disease burden and treatment disparities among certain sociodemographic groups [25].

Duvetorp et al., again by meaning of DLQI, conducted a research on the people of Sweden, Denmark and Norway noting that psoriasis alone did not have a strong impact on patients’ daily lives with regards to sports (14.5%); even if combined with psoriatic arthritis this value rose to 44% [26].

In a 2002 study by Mork et al. the authors tested the effectiveness of a variant of the classic DLQI called DLQI-N, in 230 Noerwegian Pso patients. The Authors noted that for 3 items of this questionnaire (sport, work and sexual activity) most patients considered these as “not relevant”: this, again according to the Authors, could be due to the fact that the responses “not relevant” and “no impact of the disease” on the activities in question were interpreted in the same way [27].

Van Geel et al. compared the questionnaire for adults (DLQI) and minors (Children DLQI, C-DLQI). Patients aged between 16 and 17 were sampled and both questionnaires were administered. These authors noticed a big difference between the scores concerning the “sport” item if investigated through the DLQI or the C-DLQI. The discrepancy between the two scores is believed to be due to the different definition of the item where, by adding the word “swimming” in the C-DLQI, the patients obtained a higher score. This could reflect that swimming plays an important role in adolescence not only in a competitive setting but also in other aspects of social life such as summer holidays. This study highlights, once again, the importance of the cosmetic aspect of psoriasis [28].

Leino et al. administered a specific questionnaire of 11 dichotomous categories, inserting sports and non-sports activities, to evaluate the impact of psoriasis in leisure activities. They observed that the majority of people reduced (23.7%) or completely abandoned (30.2%) sport activities because of psoriasis, with differences between sexes. While females tend to reduce sport activity, males showed a dichotomous behaviour, with complete abandonment or insignificant reduction. This may be due, according to the Authors, to the fact that males carried out these activities with greater intensity before the onset of the disease. Another important fact is that the average age of patients who decreased or abandoned sport was significantly lower than those who did not decrease it [29]. The role of Pso as limiting factor in sporting activities because of social embarrassment has been confirmed by the experience of Torres et al. [30].

In contrast with the studies so far examined, Demirel et al. suggested that mild and moderate psoriasis does not actually influence the daily physical activity of the patients: in fact, in their experience, Pso patients tend to do more physical exercise [31].

Eghlileb et al. tested the quality of life of relatives and partners of psoriatic patients. Just 8% of the subjects examined referred no impact on their quality of life depending on the disease of close Pso patients. In particular, 44% of the interviewees reported difficulties in playing sports together with relatives or partners because of the disease. To justify these limitations, elements such as hospital treatment and the embarrassment of exposing the skin in public have been brought into consideration (observation also made by the patients’ children). These results suggest an important role of the stress experienced by people alongside patients not only as a consequence of the disease but as a further reason for the aggravation of the psychological status of the patients, with possible repercussions also on the severity of the disease itself [32].

In the study of Jenner et al. inherent to economic and social burden of PSO, over one third of the patients reported difficulties in performing sports activities. One of the possible explaination is the increase of the costs related to sport activities due to a more careful choice of clothing and their cleaning, to the more specific selection of less aggressive but more expensive detergents as well as to unmanageable expenses to carry out sports in constant manner [33].

## 4. Discussion

The analysis of the Literature strongly suggests a close mutual influence between sport and Pso, showing a correlation between the quantity and/or quality of sport activities and severity and course of the disease. Sport activity, through molecular mechanisms, can reduce the levels of the pro-inflammatory cytokines systemically [6,8], with a positive influence on the chronic inflammatory state which sustains the disease. Physical activity, especially the vigorous and constant one, seems to exert a protective role rather than being an aggravating factor. It is not clear whether weight loss can be more important than a sporting activity practiced with constancy in order to improve psoriasis, even though it’s reasonable that the synergic effect of both these practices can lead to better results [8,9]. Another observation that emerges is that patients would benefit mainly from aerobic sports (swimming, tennis and running), as to emphasize that the control of blood sugar and lipid levels, through a shift of metabolism, could be crucial for a more adequate control of the disease. Sport might represent a useful non-pharmacological intervention especially in patients with cardiovascular and/or metabolic comorbities. On the other hand, these patients, regardless of their BMI, would have a lower cardiorespiratory fitness. This element should not be underestimated: if inserted in a competitive context, the earlier onset of fatigue can affect the performance of the patient with negative psychological (poor competitiveness by the athlete and loss of aims) and economic (less probability to receive endorsements to further their careers) consequences [10]. From our review, no significant differences emerge between outdoor and indoor sports. However, if we consider that heliotherapy provides strong reliefs on psoriatic symptoms, as demonstrated by several studies, we may hypothesize that performing physical activities in an outdoor setting should improve skin lesions both directly, via the anti-inflammatory effects of UV rays, and indirectly, increasing Vitamin D3 production. A few studies have focused on skin improvement through weight loss with diet alone. The loss of fat mass can reduce joint overload thus facilitating sports activity with a positive feedback mechanism. Furthermore, lowering the levels of oxidative stress due to the abundance of fat and restoring the right ratio of lean/fat mass, may facilitate physical activity. Sporting, in the end, could be considered a fundamental adjuvant in the process of slimming, especially in Pso patients often affected by concomitant obesity [15,16]. About half of the studies examined deal with QoL (10). It is well-known that Pso is considered a social stigma because of its localization in always exposed areas like face and hands, thus representing a constant source of embarrassment. Pso is also time- and money-consuming, as it requires continuous purchase and use of creams that should be applied even more times per day. This fundamental aspect of the life of each patient is investigated through special questionnaires, such as DLQI, an essential tool for the dermatologist when dealing with the patient and for the best treatment outcomes, used in most of the above reported studies. From their analysis, it emerges that a cluster of patients sharing specific characteristics (old-aged females, unemployed, without a partner) consider sports as an irrelevant aspect of their daily lives or do not see psoriasis as a limiting factor. This may result from a lack of knowledge of your own disease and/or from the devaluation of the importance of sport in the context of a healthy and balanced life. It is interesting to note that these conditions, in turn, are associated with a greater risk of cardiometabolic and psychiatric diseases and, if neglected from the beginning, can lead to an aggravation of the same conditions. For the elderly, we can hypothesize that a healthy lifestyle, including a constant physical activity before the onset of the disease, might produce better results in controlling it, especially since changing habits can prove difficult in later years [22]. It is reasonable to assume that it’s better to begin before or continue exercising after the onset of Pso: in this way, the initial embarrassment associated with skin manifestations might disappear and would not represent an obstacle when practicing outdoor sports or with partners. This point can be deduced from the study related to swimming habits among minors: it is reasonable to assume that a young patient still wants to experience various sports and has not yet established preferences. Swimming, moreover, unlike other lighter physical activities such as walking, is very often inserted in the recreational context of travel, holidays in sea locations, where one can perform other activities in the water that require the ability to swim (water polo, etc.). It is worth noting, as well, that in one study the authors researched also other parameters that can actually be object of interest in subsequent studies (such as household size and marital status) in their relation with Pso [7]. Particular subsets of Pso could be more related to the abandonment of sporting activity: for example, arthropathic or acral involvement certainly represents a physical and psychological obstacle to many sports (volleyball, basketball, etc.). The course of the disease could also be an incentive to abandonment: the sudden outburst and spread of the disease might cause feelings of shame, accelerating the forced withdrawal from training. Pso can also affect sportswear. Patients tend to adopt more covering outfits, in order to protect the affected areas of the body from sight, choosing colors to help mask any bloodstains or silvery scales [20]. A change in the times of practice of sports may be useful as to avoid the hottest hours of the day, thus preventing excessive sweating. Pso can also limit the practice of contact sports, in which the close distance between the players may expose patients to embarrassment. About psoriatic arthritis, the data of the literature are very limited, with few anecdotal reports and lack of studies on significant cohorts of patients, though the topic appears of great interest in both amateur and professional settings. Early application of non-invasive diagnostic procedure like ultrasound examination has proved to be effective in demonstrating PsA involvement of joints and tendons and is more sensitive than clinical examination in detecting the underlying pathology [34]. In this regard, instrumental screening might be proposed in sportsmen with familial and personal history of Pso to assess the risk to develop or worsen musculoskeletal damage by inappropriate physical activities, thus allowing the prosecution of a safe sporting life or addressing patients to a practice sports more safely.

## 5. Conclusions

Psoriasis has an impact on many aspects of the daily life of the patients, and the role played by sport in the quality of life seems relevant even if underestimated. There is still much to investigate about the influence of different types of sport on Pso and psoriatic arthritis at both molecular and clinical levels. In particular, the relationship between physical activity and severity of Pso in patients with cardiovascular and/or metabolic comorbidities may be of help to better understand the mechanisms underlying different ways of presentation of a unique chronic inflammatory state. Sports may represent a striking non-pharmacological resource in complex patients like the psoriatic ones, especially within programs of education and promotion of a healthier lifestyle.

## Figures and Tables

**Table 1 medicina-57-00161-t001:** The impact of sports on psoriasis.

Authors [Reference]	Year	Population	No. of Patients	Scores	Other Factors	Main Outcome (s)
Schwarz et al. [5]	2019	9940	9940	BSA	BMI	High BMI and Pso lower chances of sporting activity
Frankel et al. [6]	2012	86,655	86,655		OPEN QUESTIONNAIRE	Threshold effect of sport ameliorates Pso
Do et al. [7]	2015	6549	155		MET/min, MVPA	MET per minute lower in Pso patients than controls
Balato et al. [8]	2015	1305	400		BMI and DURATION OF SPORT	Sport prevents the development of Pso
Megna et al. [9]	2017	680	323		BMI and DURATION OF SPORT	Sport does not worsen psoriasis
Brophy et al. [10]	2008	1	1			Oligoarticular psoriatic arthritis can be worsened in case of contact sports
Corn et al. [11]	1987	1	1			Koebner’s sign is elicited by sporting activities

Abbreviations: BMI = Body Mass Index, BSA = Body Surface Area, MET = Metabolic Equivalent Task, MVPA = moderate to vigorous physical activity, Pso = psoriasis.

**Table 2 medicina-57-00161-t002:** The relevance of sports to comorbidities of psoriasis.

Authors [Reference]	Year	Population	No. of Patients	Scores	Other Factors	Main Outcome (s)
Wilson et al. [12]	2016	1093	26		MET/min	CRF lower in Pso patients than controls
Wilson et al. [13]	2017	9174	232		OPEN QUESTIONNAIRE	Pso and obesity lower the chances of slimming
Naldi et al. [14]	2014	303	303	PASI		Diet and physical exercise in Pso patient requires a full team of physicians
Kim et al. [15]	2019	9,718,591	2,595,878		BMI	Metabolic syndrome is related to development of Pso
Gyldenløve et al. [16]	2015	32	16	PASI		Insulin resistance is higher in Pso patients
Jensen et al. [17]	2016	56	56	DLQI, PASI		Weight loss improves PASI score
Phan et al. [19]	2018	35,735	3557	MEDI-LITE		Mediterranean diet protects against Pso
Ahdout et al. [20]	2012	117	65	PSS, GLTEQ, REAP, PASI		Patients with Pso have higher BMI than controls
Ferguson et al. [21]	2019	502,417	10,604		BMI	Central adiposity is linked to autoimmune disorders

Abbreviations: BMI = Body Mass Index, CRF = cardiorespiratory fitness, DLQI = Dermatology Life Quality Index, GLTEQ = Godin Leisure-Time Exercise Questionnaire, MEDI-LITE = score to measure adherence to Mediterranean diet based on the literature, MET = Metabolic Equivalent Task, PASI = Psoriasis Area Severity Index, Pso = psoriasis, PSS = perceived stress scale, REAP = Rapid Eating Assessment for Patients.

**Table 3 medicina-57-00161-t003:** The effect of psoriasis on the sport activities.

Authors [Reference]	Year	Population	No. of Patients	Scores	Other Factors	Main Outcome (s)
Ramsay et al. [22]	1988	104	104		OPEN QUESTIONNAIRE	Behavioral changes in Psoriasis >50% find Pso a social stigma
Nyunt et al. [23]	2015	223	223	DLQI, PASI		Pso limits sporting activity
Rencz et al. [24]	2018	428	428	DLQI, PASI	EQ-5D-3L	DLQI is the best way to assess quality of life
Barbieri et al. [25]	2020	1733	1733	DLQI		Unemployed/single people care less about their QoL related to Pso
Duvetorp et al. [26]	2019	1221	1221	NORPAPP		Pso and Pso arthritis have a greater combined value on QoL of patients than singularly
Mork et al. [27]	2002	230	230	DLQI-N		High percentage of “not relevant responses” in Sport
Van Geel et al. [28]	2016	56	56	DLQI, CDLQI		DLQI and C-DLQI provide different results in the same teenager patients
Leino et al. [29]	2014	262	262	VAS		>50% Pso patients reduce/abandon sport
Torres et al. [30]	2014	250	90	IPAQ-S		Levels of physical activity are lower in Pso patients
Demirel et al. [31]	2013	60	30	DLQI	FITNESS APTNESS	Pso does not affect sporting life
Eghlileb et al. [32]	2007	63	63	PDI, PASI, DLQI		Quality of life in patients’ relatives is worse in 44% of cases
Jenner et al. [33]	2002	83	83	PDI, PASI, SAPASI		Cost of Sports in psoriatic patients >33% find Pso a limit to sporting activity

Abbreviations: CDLQI = children DLQI, DLQI-N = norvegian version of DLQI, EQ-5D-3L = third level of EuroQol five dimension questionnaire, IPAQ-S = International Physical Activity Questionnaire short form, NORPAPP = NORdic PAtient survey of Psoriasis and Psoriatic arthritis, PASI = Psoriasis Area Severity Index, PDI = pain disability index; Pso = psoriasis, QoL = quality of life, SAPASI = Self-administered PASI, VAS = visual analog scale.

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
