# Peer review of "Mutual Influence of Psoriasis and Sport"

_medicina, 2021, doi:10.3390/medicina57020161_

Round 1

Reviewer 1 Report

Great and thorough reivew paper highlighting underappreciated and under-researched topic concerning the relationship between sport and psoriasis. Well done.

The authors conducted a thorough research using Pubmed database in order to elucidate the relationship between sport and psoriasis. The topic has been poorly investigated so far thus the paper is very relevant and original and points to many unsolved questions dealing with the role of psoriasis in influencing the sporting habits of patients and the other way its impact on the course of the disease. The authors thoroughly  conducted a more discursive review than previously published papers. The text is well written and easy to read. The conclusions are consistent with the evidence and arguments presented and they address the main question posed. The authors indeed concluded that psoriasis has an impact on many aspects of the daily life including sport habbits and they raised that the role played by sport in the quality of patients seems relevant or even underestimated. They also pointed that there is still much to investigate about the influence of different types of sport on psoriasis at both molecular and clinical levels and highlighted the importance of sports in non-pharmacological resource in complex antipsoriatic management.

Author Response

We thank the reviewer for the positive comments.

Reviewer 2 Report

In the paper entitled "Mutual influence of psoriasis and sport" Paolo et al. reviewed the effects of physical activity on the psoriasis. Several points should be corrected in this manuscript.

  1. In the results section, to better understand what you're trying to say, you need to put together important things. It is needed to be largely sorted out in order as following, for example 1) the impact of sports on psoriasis, 2) the effect of psoriasis on the sport activities, 3) the relevance of the sports to comorbidities of psoriasis.
  2. In Table 1, the conclusion or main results of the references should be added in main outcome part, respectively.

Author Response

Q1. In the results section, to better understand what you're trying to say, you need to put together important things. It is needed to be largely sorted out in order as following, for example 1) the impact of sports on psoriasis, 2) the effect of psoriasis on the sport activities, 3) the relevance of the sports to comorbidities of psoriasis.

A1. We modified the result section according to the reviewer’s suggestions. We created three new subcategories, respectively: 1) the impact of sports on psoriasis, 2) the effect of psoriasis on the sport activities, 3) the relevance of the sports to comorbidities of psoriasis. We resumed the main topic of each specific subcategory and then delved deeper in the specific articles. We moved some of the topics to better suit the new subcategories. The reference order was therefore changed. Small typing mistakes were corrected. Here are the changes, as follow:

1)Page 2, Line 117-120: 3.1.The impact of sports on psoriasis

Sporting activity has several effects on the Pso patients: it can ameliorate the lesions, leave them as they are or worsen them.

2)Page 3, Line 136-148A relevant question about the possible side effects of sport in case of psoriatic disease was posed by Brophy et al. In a case report, they describe the case of a 39-year-old athlete who, presenting with recurrent monolateral knee effusions, received a late diagnosis of psoriatic arthritis since his injuries were initially diagnosed as due to trauma while playing; in addition, the patient had a positive family history of seronegative and non-seronegative spondyloarthritis. Following a suitable treatment, the patient not just recovered the complete function of his knee, but also noticed improvements in other joints which previously had not prevented him from attending competitive sports. As the Authors say, a patient suffering from oligoarticular forms of psoriatic arthritis not only would benefit from an early and aggressive treatment, but also should be aware of an intrinsic structural weakness of his own cartilages, and therefore limit the use of the joints in sports like the contact ones. We see, therefore, how sport presents itself both as a potential target for the treatment of psoriatic disease and, on the other hand, as a source of possible injury [10].

A similar event was reported by Corn et al., concerning a patient known for a history of psoriasis who had seen the appearance of characteristic lesions on the soles of the feet following sports, seeing how the Koebner phenomenon could, therefore, be a spy of a pathology otherwise with minimal clinical manifestations [11].

3)Page 4, Line 157-161: 3.2 The relevance of sports to comorbidities of psoriasis

Psoriasis can have several comorbidities linked to it, mostly dismetabolic ones caused by lack of exercise. One of the main means to evaluate a patient’s fitness is through examination of the cardiorespiratory fitness (CRF) The link between CRF

4)Page 5, Line 216-221: Finally, another study by Ferguson et al. suggests a role of central adiposity in the development of psoriasis, psoriatic arthritis and rheumatoid arthritis: all of these conditions were significantly associated with overweight and lack of physical activity, even after correction of the BMI, thus suggesting that the weight could be somehow linked to the development of several autoimmune diseases [21].

3.3 the effect of psoriasis on the sport activities

Q.2 In Table 1, the conclusion or main results of the references should be added in main outcome part, respectively.

A2. The requested outcomes were added into the revised version of the manuscript for a more immediate consultation. Here as follows:

1)Page 8-9, Line 459-639:

Koebner’s sign is elicited by sporting activities

Behavioral changes in Psoriasis>50% find Pso a social stigma

Cost of Sports in psoriatic patients>33% find Pso a limit to sporting activity

High percentage of “not relevant responses” in Sport

Quality of life in patients’ relatives is worse in 44% of cases

Oligoarticular psoriatic arthritis can be worsened in case of contact sports

Patients with Pso have higher BMI than controls

Threshold effect of sport ameliorates Pso

Pso does not affect sporting life

>50% Pso patients reduce/abandon sport

Diet and physical exercise in Pso patient requires a full team of physicians

Levels of physical activity are lower in Pso patients

Sport prevents the development of Pso

MET per minute lower in Pso patients than controls

Insulin resistance is higher in Pso patients

Pso limits sporting activity

Weight loss improves PASI score

DLQI and C-DLQI provide different results in the same teenager patients

CRF lower in Pso patients than controls

Sport does not worsen psoriasis

Pso and obesity lower the chances of slimming

Mediterranean diet protects against Pso

DLQI is the best way to assess quality of life

Pso and Pso arthritis have a greater combined value on QoL of patients than singularly

Central adiposity is linked to autoimmune disorders

Metabolic syndrome is related to development of Pso

High BMI and Pso lower chances of sporting activity

Unemployed/single people care less about their QoL related to Pso

Round 2

Reviewer 2 Report

Table 1 contains all related reference articles. Divide Table 1. into three tables related to 1) the impact of sports on psoriasis, 2) the effect of psoriasis on the sport activities, 3) the relevance of the sports to comorbidities of psoriasis.

Author Response

Q: Table 1 contains all related reference articles. Divide Table 1. into three tables related to 1) the impact of sports on psoriasis, 2) the effect of psoriasis on the sport activities, 3) the relevance of the sports to comorbidities of psoriasis.

A: Table 1 has been divided into three different tables, one for each item suggested. Corrections are listed from line 460 of page 7 through line 513 of page 9.